# Olaparib Conjugates with Selenopheno[3,2-*c*]quinolinone Inhibit PARP1 and Reverse ABCB1-Related Multidrug Resistance

**DOI:** 10.3390/pharmaceutics14122571

**Published:** 2022-11-23

**Authors:** Marina Makrecka-Kuka, Jelena Vasiljeva, Pavels Dimitrijevs, Pavel Arsenyan

**Affiliations:** Latvian Institute of Organic Synthesis, Aizkraukles 21, LV-1006 Riga, Latvia

**Keywords:** cancer, MDR, olaparib, quinolinone, PARP1, selenium

## Abstract

The restoration of the efficacy of antitumor medicines is a cornerstone in the combat with multidrug resistant (MDR) cancers. The overexpression of the ABCB1 transporter is a major obstacle to conventional doxorubicin therapy. The synergy of ABCB1 suppression and PARP1 activity inhibition that hampers malignant cell DNA repair could be a powerful tool in anticancer therapy. Herein, we report the design and synthesis of three novel olaparib conjugates with selenophenoquinolinones, their ability to reverse doxorubicin resistance in uterus sarcoma cells as well as their mechanism of action. It was found that the most potent chemosensitizer among studied compounds preserves PARP1 inhibitory activity and attenuates cells’ resistance to doxorubicin by inhibiting ABCB1 transporter activity. These results demonstrate that the conjugation of PARP inhibitors with selenophenoquinolinones is a prospective direction for the development of agents for the treatment of MDR cancers.

## 1. Introduction

In recent years, PARP inhibitors (PARPi) have become a standard of care in various cancers with BRCA1 or BRCA2 mutations [1,2,3,4,5]. Among these medications, olaparib is the most used drug in monotherapy or in combination with other chemotherapy agents [6]. Unfortunately, ABCB1-mediated resistance is the reason for the major failure of ovarian cancer treatment with paclitaxel, doxorubicin, and PARPi, such as olaparib or rucaparib [7,8,9]. Rottenberg et al. showed that, despite good initial response in the *Brca1*^−/−^;*p53*^−/−^ mouse model, prolonged treatment with olaparib eventually led to resistance development in all tumors. Strikingly, they confirmed that the expression of *Abcb1a* and/or *Abcb1b* genes was increased by 2 to 85 times in 11 of 15 olaparib-resistant tumors [10]. In turn, Lombard et al. demonstrated that upregulated ABCB1 mediates cross-resistance between taxanes and olaparib in C4-2B prostate cancer cells, which can be overcome by decreasing ABCB1 expression or inhibiting ABCB1 using elacridar or enzalutamide [11]. A phase II clinical trial of cediranib with olaparib in the patients with progression of ovarian cancer after PARPi treatment showed that ABCB1 upregulation is clinically important, and a poor outcome is observed if ABCB1 expression is increased [12]. Although verapamil and the third generation ABCB1 inhibitors such as zosuquidar, elacridar, and tariquidar reduce resistance to olaparib and have shown promising results in preclinical studies, these agents have not been approved for cancer treatment so far due to the high risk of serious adverse effects or lack of clinical benefit [13,14,15].

In addition, ABCB1 overexpression is a major contributor to tumor resistance to doxorubicin, another substrate of the transporter [16,17,18]. Synergy in the anticancer activity of olaparib and doxorubicin was established in cell cultures and mouse xenograft models of osteosarcoma [19], as well as in phase II clinical trials for platinum-resistant ovarian cancer regardless of BRCA status [20]. Since both olaparib and doxorubicin are substrates for the ABCB1 transporter, we hypothesized that a drug conjugate combining PARP and ABCB1 inhibitory activity along with doxorubicin co-treatment could cause a synergistic triangle to come into being.

Selenium-containing compounds have already proven themselves as potential MDR reversing agents [21,22,23,24]. Recently, continuing our research of antitumor agents [25,26], we reported a novel selenopheno[3,2-c]quinolinone that attenuates cells’ resistance to doxorubicin by inhibiting ABCB1 transporter activity [27]. Here, we would like to present our progress in the development of conjugates consisting of selenopheno[3,2-*c*]quinolinone and olaparib pharmacophore moieties linked through dicarboxylic acids (Figure 1).

## 2. Materials and Methods

### 2.1. Experimental

Unless otherwise stated, all reagents were purchased from commercial suppliers and used without further purification. Thin layer chromatography (TLC) was performed using Merck silica gel 60 F254 plates and visualized by UV (254 nm) fluorescence. ZEOCHEM silica gel (ZEOprep 60/35–70 microns–SI23501) was used for column chromatography. ^1^H, ^13^C, and ^77^Se NMR spectra were recorded on a Bruker Avance Neo spectrometer at 400, 101, and 76 MHz correspondingly at 303 K in CDCl_3_/TMS, CD_3_OD, or DMSO-d_6_ solution. The ^1^H chemical shifts are given relative to TMS, ^13^C–relative to CDCl_3_, CD_3_OD or DMSO, and ^77^Se–relative to dimethyl selenide. The melting points were determined on a Optimelt, and the results are given without correction. LC-MS spectra were recorded on Waters 3100 Mass Detector Acquity UPLC. HRMS spectra were recorded on a Waters Synapt GII Q-ToF UPLC/MS system. HPLC spectra were recorded on Waters Alliance separation modules with a UV/VIS detector.

### 2.2. General Procedure for the Preparation of Selenopheno[3,2-c]quinolinones ***2**–**4***

In a round-bottom flask, HBTU (238 mg, 0.63 mmol, 1.1 equiv), dicarboxylic acid (1.71 mmol, 3.0 equiv), and DIPEA (596 μL, 3.42 mmol, 6.0 equiv) were dissolved in DMF (2 mL), and then the mixture was stirred at room temperature for 5 min. Compound **1** (300 mg, 0.57 mmol, 1.0 equiv) in DMF was added to the mixture, and the resulting mixture was stirred at room temperature for 16 h. The solvent was then evaporated under the flow of the air, and the crude residue was purified by flash column chromatography with DCM/MeOH in gradient to obtain a mixture of the product **2**–**4** and dicarboxylic acid. Dicarboxylic acid was then removed by reverse phase flash column chromatography with MeCN/H_2_O.

#### 2.2.1. 4-(4-((3-Bromo-5-methyl-4-oxo-4,5-dihydroselenopheno[3,2-c]quinolin-2-yl) methyl)-1,4-diazepan-1-yl)-4-oxobutanoic Acid (**2**)

Yield 0.19 g (61%), white solid, mp >200 °C. ^1^H NMR (400 MHz, DMSO-d_6_) δ 7.80–7.70 (m, 1H, H-6), 7.61–7.53 (m, 1H, H-7), 7.53–7.47 (m, 1H, H-9), 7.28–7.14 (m, 1H, H-8), 3.96–3.81 (m, 2H, CH_2_), 3.61 (s, 3H, CH_3_), 3.58–3.48 (m, 4H, CH_2_), 2.91–2.85 (m, 1H, CH_2_), 2.81–2.67 (m, 3H, CH_2_), 2.60–2.53 (m, 2H, CH_2_), 2.49–2.41 (m, 2H, CH_2_), 1.90–1.67 (m, 2H, CH_2_). ^13^C NMR (101 MHz, DMSO-d_6_) δ 174.0, 170.6, 170.5, 156.6, 149.3, 147.8, 147.7, 136.3, 130.3, 127.7, 125.2, 122.5, 118.4, 115.4, 105.0, 105.0, 57.7, 56.0, 55.7, 55.0, 54.5, 47.3, 45.8, 45.1, 43.7, 29.3, 29.1, 27.9, 27.7, 27.6, 27.1. ^77^Se NMR (76 MHz, DMSO-d_6_) δ 601.9. Calculated for C_22_H_24_BrN_3_O_4_Se: *m*/*z*: 554.0188, found, *m*/*z*: 554.0195 [M+H]^+^. HPLC: 99.2% (RT = 6.96 min, Apollo C18-12 (4.6 mm × 150 mm), mobile phase 5–95% acetonitrile + 0.1% H_3_PO_4_, 1 mL min^−1^, 40 °C).

#### 2.2.2. 10-(4-((3-Bromo-5-methyl-4-oxo-4,5-dihydroselenopheno[3,2-c]quinolin-2-yl) methyl)-1,4-diazepan-1-yl)-10-oxodecanoic Acid (**3**)

Yield 0.12 g (35%), foam. ^1^H NMR (400 MHz, CD_3_OD) δ 7.49–7.37 (m, 2H, H-6,7), 7.35–7.28 (m, 1H, H-9), 7.14–7.07 (m, 1H, H-8), 3.84 (d, *J* = 7.0 Hz, 2H, CH_2_), 3.69–3.59 (m, 4H, CH_2_), 3.57 (d, *J* = 2.5 Hz, 3H, CH_3_), 2.96–2.82 (m, 2H, CH_2_), 2.82–2.73 (m, 2H, CH_2_), 2.46–2.34 (m, 2H, CH_2_), 2.30–2.17 (m, 2H, CH_2_), 2.00–1.79 (m, 2H, CH_2_), 1.68–1.47 (m, 4H, CH_2_), 1.43–1.22 (m, 8H, CH_2_). ^13^C NMR (101 MHz, CD_3_OD) δ 177.5, 175.3, 158.8, 151.5, 148.5, 148.1, 137.4, 131.3, 128.9, 126.0, 123.9, 120.2, 116.3, 106.7, 59.9, 59.7, 57.6, 56.9, 56.9, 56.1, 46.8, 45.7, 34.9, 34.3, 34.1, 30.5, 30.4, 30.3, 30.2, 30.2, 29.9, 28.8, 26.6, 26.1. ^77^Se NMR (76 MHz, CD_3_OD) δ 598.4. Calculated for C_28_H_36_BrN_3_O_4_Se: *m*/*z*: 638.1127, found, *m*/*z*: 638.1131 [M+H]^+^. HPLC: 98.6% (RT = 9.25 min, Apollo C18-12 (4.6 mm × 150 mm), mobile phase 5–95% acetonitrile + 0.1% H_3_PO_4_, 1 mL min^−1^, 40 °C).

#### 2.2.3. 12-(4-((3-Bromo-5-methyl-4-oxo-4,5-dihydroselenopheno[3,2-c]quinolin-2-yl) methyl)-1,4-diazepan-1-yl)-12-oxododecanoic Acid (**4**)

Yield 0.11 g (29%), foam. ^1^H NMR (400 MHz, CDCl_3_) δ 7.65–7.53 (m, 2H, H-6,7), 7.42–7.32 (m, 1H, H-9), 7.29–7.21 (m, 1H, H-8), 4.74 (s, 2H, CH_2_), 4.02–3.79 (m, 2H, CH_2_), 3.72 (s, 3H, CH_3_), 3.67–3.57 (m, 2H, CH_2_), 3.48–3.22 (m, 4H, CH_2_), 2.42–2.19 (m, 6H, CH_2_), 1.69–1.49 (m, 4H, CH_2_), 1.38–1.16 (m, 12H, CH_2_). ^13^C NMR (101 MHz, CDCl_3_) δ 177.9, 174.1, 173.7, 157.7, 154.8, 137.1, 131.6, 128.7, 127.1, 125.8, 123.2, 118.6, 117.8, 115.2, 55.9, 55.6, 53.8, 52.0, 50.9, 45.6, 43.6, 42.6, 40.4, 34.0, 33.6, 29.9, 25.2, 24.9, 24.8, 24.1, 23.2. ^77^Se NMR (76 MHz, CDCl_3_) δ 625.1. Calculated for C_30_H_40_BrN_3_O_4_Se: *m*/*z*: 666.1440, found, *m*/*z*: 666.1453 [M+H]^+^. HPLC: 96.0% (RT = 10.51 min, Apollo C18-12 (4.6 mm × 150 mm), mobile phase 5–95% acetonitrile + 0.1% H_3_PO_4_, 1 mL min^−1^, 40 °C).

### 2.3. General Procedure for the Preparation of Selenopheno[3,2-c]quinolinones ***5a**–**c***

In a round-bottom flask, carboxylic acid **2**–**4** (100 mg, 1.0 equiv), 4-(4-Fluoro-3-(piperazine-1-carbonyl)benzyl)phthalazin-1(2*H*)-one (1.3 equiv), hydroxybenzotriazole (1.5 equiv), and 1-ethyl-3-carbodiimide hydrochloride (3 equiv) were dissolved in DMF (2 mL), and *N*-methylmorpholine (3 equiv) was added. Mixture was stirred at room temperature for 3–4 h. The solvent was then evaporated under the flow of the air, and the crude residue was purified by flash column chromatography (for compounds **5a**,**b**, eluent: MeCN/H_2_O in gradient, for compound **5c** eluent: DCM/EtOAc/MeOH in ratio 1/1/0.05).

#### 2.3.1. 1-(4-((3-Bromo-5-methyl-4-oxo-4,5-dihydroselenopheno[3,2-c]quinolin-2-yl) methyl)-1,4-diazepan-1-yl)-4-(4-(2-fluoro-5-((4-oxo-3,4-dihydrophthalazin-1-yl)methyl)benzoyl)piperazin-1-yl)butane-1,4-dione (**5a**)

Yield 0.12 g (79%), white solid, mp >200 °C. **5a** appears in spectra as signals of 2 rotamers. ^1^H NMR (400 MHz, DMSO-*d*_6_ + two drops of HCl) δ 12.60 (s, 1H, NH), 8.32–8.15 (m, 1H, Ar), 8.01–7.93 (m, 1H, Ar), 7.93–7.86 (m, 2H, Ar), 7.86–7.79 (m, 1H, Ar), 7.72–7.42 (m, 3H, Ar), 7.37–7.17 (m, 3H, Ar), 4.86–4.57 (m, 2H, CH_2_), 4.33 (s, 2H, CH_2_), 4.26–3.94 (m, 1H, CH_2_), 3.60 (m, 16H, CH_2_, CH_3_), 3.24–2.98 (m, 4H, CH_2_), 2.72–2.58 (m, 2H, CH_2_), 2.42–1.98 (m, 2H, CH_2_). ^13^C NMR (101 MHz, CD_3_OD/CDCl_3_) δ 172.8, 171.0, 170.9, 166.4 (d, J = 12.2 Hz), 160.7, 158.0, 157.7, 155.6, 155.0, 145.7, 136.8, 134.5, 133.6, 131.9, 131.9, 131.5, 131.4, 129.4, 128.9 (dd, J = 2.9 Hz, J = 12.2 Hz), 128.8, 128.2, 128.1, 127.9, 126.6, 125.7, 125.0, 123.1, 55.8, 118.5, 116.1 (dd, J = 7.2 Hz, J = 22.2 Hz), 115.1 (d, J = 7.2 Hz), 54.7, 54.6, 51.8, 51.5, 46.7, 46.5, 45.3, 45.2, 44.7, 42.8, 41.9, 41.7, 41.1, 40.2, 37.3, 29.5, 28.0, 27.5, 23.9. ^19^F NMR (376 MHz, CD_3_OD/CDCl_3_) δ -118.2. ^77^Se NMR (76 MHz, CD_3_OD/CDCl_3_) δ 621.4. Calculated for C_42_H_41_BrFN_7_O_5_Se: *m*/*z*: 902.1575, found, *m*/*z*: 902.1556 [M+H]^+^. HPLC: 98.3% (RT = 7.95 min, Apollo C18-12 (4.6 mm × 150 mm), mobile phase 5–95% acetonitrile + 0.1% H_3_PO_4_, 1 mL min^−1^, 40 °C).

#### 2.3.2. 1-(4-((3-Bromo-5-methyl-4-oxo-4,5-dihydroselenopheno[3,2-c]quinolin-2-yl) methyl)-1,4-diazepan-1-yl)-10-(4-(2-fluoro-5-((4-oxo-3,4-dihydrophthalazin-1-yl)methyl)benzoyl)piperazin-1-yl)decane-1,10-dione (**5b**)

Yield 75 mg (49%), light yellow solid, mp = 172–174 °C. **5b** appears in spectra as signals of 2 rotamers. ^1^H NMR (400 MHz, DMSO-*d*_6_ + 2 drops of HCl) δ 12.59 (s, 1H, NH), 8.25 (dd, *J* = 7.8, 1.1 Hz, 1H, Ar), 7.98–7.85 (m, 3H, Ar), 7.85–7.79 (m, 1H, Ar), 7.71–7.62 (m, 1H, Ar), 7.63–7.55 (m, 1H, Ar), 7.48–7.39 (m, 1H, Ar), 7.39–7.28 (m, 2H, Ar), 7.27–7.18 (m, 1H, Ar), 4.74 (s, 2H, CH_2_), 4.32 (s, 2H, CH_2_), 3.66 (s, 3H, CH_3_), 3.63–3.44 (m, 10H, CH_2_), 3.21–3.10 (m, 2H, CH_2_), 2.89 (s, 2H, CH_2_), 2.73 (s, 2H, CH_2_), 2.38–2.19 (m, 5H, CH_2_), 2.20–1.96 (m, 1H, CH_2_), 1.56–1.38 (m, 4H, CH_2_), 1.32–1.16 (m, 8H, CH_2_). ^13^C NMR (101 MHz, DMSO-*d*_6_) δ 172.2, 170.9, 164.0, 162.3, 159.4, 157.6, 156.6, 155.1, 144.8, 136.8, 134.8, 133.5, 131.7 (d, J = 8.5 Hz), 131.6, 129.1, 128.9 (d, J = 4.1 Hz), 127.9, 126.1, 125.7, 125.5, 123.7, 123.5, 122.9, 117.9, 116.0, 115.8, 115.7, 46.7, 46.3, 44.9, 44.5, 41.7, 41.3, 41.1, 40.5, 36.4, 35.8, 32.6, 32.5, 32.6, 30.8, 29.5, 28.8, 28.7. ^19^F NMR (376 MHz, DMSO-*d*_6_) δ-118.2 (m). ^77^Se NMR (76 MHz, DMSO-*d*_6_) δ 626.0. Calculated for C_48_H_53_BrFN_7_O_5_Se: *m*/*z*: 986.2514, found, *m*/*z*: 986.2501 [M+H]^+^. HPLC: 98.6% (RT = 7.80 min, Apollo C18-12 (4.6 mm × 150 mm), mobile phase 5–95% acetonitrile + 0.1% H_3_PO_4_, 1 mL min^−1^, 40 °C).

#### 2.3.3. 1-(4-((3-Bromo-5-methyl-4-oxo-4,5-dihydroselenopheno[3,2-c]quinolin-2-yl) methyl)-1,4-diazepan-1-yl)-12-(4-(2-fluoro-5-((4-oxo-3,4-dihydrophthalazin-1-yl)methyl)benzoyl)piperazin-1-yl)dodecane-1,12-dione (**5c**)

Yield 51 mg (34%), light yellow oil. **5c** appears in spectra as signals of 2 rotamers. ^1^H NMR (400 MHz, CDCl_3_) δ 11.33 and 11.26 (s of two rotamers, 1H), 8.50–8.32 (m, 1H), 7.78–7.63 (m, 3H), 7.63–7.56 (m, 1H), 7.54–7.46 (m, 1H), 7.38–7.27 (m, 3H), 7.24–7.15 (m, 1H), 7.06–6.95 (m, 1H), 4.27 (s, 2H), 3.92 and 3.89 (s of two rotamers, 2H, CH_2_), 3.82–3.61 (m, 4H), 3.72 and 3.71 (s of two rotamers, 3H, CH_3_), 3.60–3.47 (m, 4H), 3.45–3.15 (m, 3H), 2.93–2.70 (m, 4H), 2.42–2.19 (m, 4H), 2.16–1.96 (m, 1H), 1.96–1.79 (m, 2H), 1.70–1.48 (m, 4H), 1.38–1.10 (m, 12H), 0.91–0.71 (m, 3H). ^13^C NMR (101 MHz, CDCl_3_) δ 173.0, 172.9, 172.2, 171.9, 165.3, 165.1, 160.7, 158.2, 158.0, 155.8, 150.1, 146.6 (d, J = 16.0 Hz), 145.5, 136.8, 134.5 (d, J = 3.4 Hz), 133.6, 131.7 (d, J = 8.0 Hz), 131.6, 130.1 (d, J = 8.0 Hz), 129.6, 129.4, 129.2, 128.6, 128.4, 127.2, 125.3, 125.0, 123.9-123.5 (m), 122.5 (d, J = 5.0 Hz), 119.4 (d, J = 3.0 Hz), 116.4-116.1 (m), 115.0 (d, J = 3.0 Hz), 106.7, 106.6, 58.9 (d, J = 8.0 Hz), 57.0, 56.7, 56.2, 55.0, 48.3, 47.2, 46.9, 45.9, 45.2, 44.5, 42.3, 42.1, 41.7, 41.2, 37.8, 33.6, 33.5, 33.3, 29.7, 29.5, 28.8, 27.7, 25.4, 25.2. ^19^F NMR (376 MHz, CDCl_3_) δ -117.6–117.7 (m). ^77^Se NMR (76 MHz, CDCl_3_) δ 596.1 and 595.5. Calculated for C_50_H_57_BrFN_7_O_5_Se: *m*/*z*: 1014.2827 [M+H]^+^, found, *m*/*z*: 1014.2866. HPLC: 97.4% (RT = 6.09 min, Apollo C18–12 (4.6 mm × 150 mm), mobile phase 5–95% acetonitrile + 0.1% H_3_PO_4_, 1 mL min^−1^, 40 °C).

### 2.4. Cytotoxicity Assay

MES-SA (Human uterine sarcoma, ATCC^®^ CRL-1976™) and H9C2 (rat cardiomyocytes, ATCC CRL-1446™), MCF-7 (human adenocarcinoma, ATCC HTB-22™), and HCC1937 (human breast carcinoma, ATCC CRL-2336™) cell lines were obtained from American Type Culture Collection, and the MES-SA/Dx-5 (doxorubicin resistant human uterine sarcoma with high levels of MDR1 mRNA and P-glycoprotein, ECACC 95051031-1VL) cell line was obtained from the European Collection of Authenticated Cell Cultures. Cells were cultured in modified McCoy’s 5a medium supplemented with 10% fetal bovine serum. The cells were cultivated in a 37 °C with 5% CO_2_, 95% air, and complete humidity. After reaching approx. 90% confluence, cells were detached using 0.05% trypsin/EDTA solution and counted. Then, cells were plated at optimal density for the logarithmic phase of growth. MES-SA cells were seeded onto 96-well plate at a concentration of 3000 cells per well. Blank control wells were left cell-free for background absorption measurement. Cells had been incubated for 24 h to allow cells to adhere to the bottom of wells; then, serial dilutions of the test compounds in the medium were prepared and added to the cells (*n* = 6). Control cells were incubated with media without compounds. MTT assays were performed after 48 h incubation. Briefly, culture medium was removed from each well and replaced with fresh medium with MTT (0.2 mg/mL). After 3 h, the MTT solution was removed and replaced with 200 µL of DMSO and 25 µL Sorenson’s glycine buffer (glycine 0.1M, NaCl 0.1M, pH = 10.5 with 0.1M NaOH). The plate was further shaken for 15 min at room temperature, and the optical density at 540 nm of the wells was measured using multichannel spectrophotometer (Thermo Scientific Multiskan EX, Waltham, MA, USA). The percent of alive cells were calculated according to the formula 100∙T/C, where T is the optical density of a test well after 48 h exposure to compound, and C is control cells optical density after 48 h. All compounds were tested in three independent experiments. The IC_50_ values were calculated using Graph Pad Prism (GraphPad, Inc., La Jolla, CA, USA).

To calculate reversal fold (RF) values, doxorubicin cytotoxicity was determined in the presence of the tested compounds at their corresponding IC_20_ concentration as described above. RF values were calculated as IC_50_ of DOX/IC_50_ of DOX in the presence of a tested compound.

### 2.5. Rhodamine-123 Accumulation Assay

Inhibitory activity of the compounds on ABCB1 transporter was evaluated using a rhodamine-123 (Rho-123) accumulation assay, a specific substrate for ABCB1. MES-SA/Dx-5 and MES-SA were seeded in 12-well plates at a density of 15 × 10^4^ cells per well and cultured overnight. The cells were preincubated with **5a**–**c** (1, 2, 5, 10 µM) for 30 min, then treated with Rho-123 at a final concentration of 5.4 µM. After another 30 min of incubation, the accumulation of Rho-123 was stopped by the addition of ice-cold PBS. Cells were collected, and intracellular Rho-123 fluorescence intensity was analyzed by flow cytometry (BD FACSMelody^TM^, BD Biosciences, San Jose, CA, USA). The obtained data were analyzed using FlowJo software.

### 2.6. Statistical Analysis

Data are presented as the mean ± SD. An unpaired two-sample Student’s *t*-test was used to compare treated samples with control samples. Normal distribution was assessed using a Shapiro–Wilk test. The differences were considered significant when *p* < 0.05. The data were analyzed using GraphPad Prism (GraphPad, Inc., La Jolla, CA, USA).

## 3. Results and Discussion

It has been shown that substantial modification or replacement of the piperazine ring in the olaparib molecule can significantly reduce its DNA damaging properties [28]; however, modification at N-terminus of the piperazine is possible [29,30]. The synthesis of the proposed conjugates **5a**–**c** (Figure 1) was started from selenophenoquinolinone **1** [27]. It was treated with an excess of dicarboxylic acid (*n* = 2, 8, 10) in the presence of HBTU and DIPEA. Monoacylation products **2**–**4** were isolated by flash column chromatography using the mixture of dichloromethane/methanol in gradient (0–8%) to obtain a mixture of the product and residual dicarboxylic acid. Then, dicarboxylic acid was removed by reverse phase flash column chromatography. Compounds **2**–**4** were isolated in moderate yields (29–61%). Next, the subsequent coupling reaction of 4-(4-fluoro-3-(piperazine-1-carbonyl)benzyl)phthalazin-1(2*H*)-one with **2**–**4** led to the formation of the desired conjugates **5a**–**c** (Appendix A).

Next, PARP1 inhibition activity of the three new conjugates **5a–c** was evaluated (Table 1, Figure 2A), and olaparib was used as a reference compound. According to the obtained data, conjugate **5a** exhibits almost the same inhibiting activity as olaparib (IC_50_ = 4.6 and 3.5 nM, respectively). The elongation of the aliphatic chain between selenophenoquinolinone moiety and olaparib pharmacophore up to eight carbon atoms led to a slight decrease in activity (IC_50_ = 13.0 nM, **5b**). However, further extension of the linker alkyl chain length resulted in a substantial loss of activity (IC_50_ = 147 nM, **5c**). Notably, selenophenoquinolinone **1** did not decrease PARP1 activity even at 1 µM concentration confirming the fact that olaparib moiety in conjugates **5a**–**c** is responsible for the inhibition of PARP1. The cytotoxicity of **5a**–**c** is summarized in Table 1.

Olaparib had no cytotoxic effect on all studied cell lines (IC_50_ >100 µM). In addition, **1** showed medium cytotoxicity (IC_50_ in the range of 17 to 21 µM) on uterus sarcoma (MES-SA) and carcinoma cells (MCF-7 and HCC1397). It should be noted that **1** as well as all studied conjugates **5a**–**c** are not toxic to rat cardiomyoblasts H9C2 (IC_50_ >300 µM). The cytotoxic profile of **5a** clearly indicates the capacity to suppress growth of uterus sarcoma cells (MES-SA, IC_50_ = 7.1 ± 2.7 µM) and BRCA1-deficient breast carcinoma HCC1937 cells (IC_50_ = 3.1 ± 1.0 µM). The elongation of a linker between selenophenoquinolinone and olaparib fragment led to the disappearance of the cytotoxic effect.

Next, the cytotoxicity of **5a**–**c** was evaluated on a doxorubicin resistant human uterus sarcoma cell line (MES-SA/Dx-5) using the MTT assay (Table 2). IC_50_ value of doxorubicin was approx. 28 times higher compared to non-resistant MES-SA cells (Table 2), confirming the resistance of the cells to doxorubicin. Surprisingly, compound **5b** was almost 20 times more cytotoxic to MES-SA/Dx-5 than regular MES-SA cells (IC_50_ = 1.667 ± 0.03 μM and 32.5 ± 1.6 μM, respectively), while other conjugates exhibited moderate cytotoxicity on MES-SA/Dx-5 cells. To explore the potential of the novel compounds to reverse doxorubicin resistance, cytotoxicity of doxorubicin was evaluated in the presence of **5a**–**c** at their corresponding IC_20_ concentrations. Reversal Fold (RF) values were calculated according to Equation (1):(1)$$\frac{{\mathrm{IC}}_{50}\;\mathrm{of}\;\mathrm{DOX}}{{\mathrm{IC}}_{50}{\;\mathrm{of}\;\mathrm{DOX}\;\mathrm{with}\;a\;\mathrm{conjugate}\;\mathrm{at}\;\mathrm{IC}}_{20}}$$

As ABCB1 is overexpressed in doxorubicin-resistant tumor cells, we explored the effect of **5a**–**c** on the function of ABCB1-mediated efflux. To estimate the function of the ABCB1 transporter, intracellular accumulation of Rhodamine-123 (Rho-123), a fluorescent substrate of ABCB1, was measured by fluorescence activated cell sorting (FACS). According to the obtained results, only pretreatment with **5b** increased intracellular Rho-123 fluorescence intensity in MES-SA/Dx-5 cells (Figure 2B,D,E). After 30 min incubation with 10 µM **5b,** 26.1% of cells were Rho-123 positive, compared to 19.7% of cells in the control experiment (*p* = 0.0054, Student’s *t*-test). Notably, **5b** increased Rho-123 positive cell population in a dose-dependent manner (Figure 2C,F,G). Statistically significant differences between control and treated cells can be observed starting from 2 µM of **5b**. Therefore, acute treatment with **5b** inhibits the activity of ABCB1 transporter, thus decreasing the efflux of doxorubicin and consequently reversing drug resistance. However, **5c** showed the ability to reverse MES-SA/Dx5 cells’ resistance to doxorubicin, and it is clear that the mechanism is not related to ABCB1 activity.

To sum up, conjugation of the selenopheno[3,2-*c*]quinolinone pharmacophore with olaparib through dicarboxylic acids preserves PARP1 inhibitory activity; however, activity decreases with elongation of the linker alkyl chain. Derivative **5b** effectively suppresses resistant uterus sarcoma MES-SA/Dx5 cell growth and, at the same time, is harmless to cardiomyoblasts. Compound **5b** is a potent PARP1/ABCB1 inhibitor and exhibits a synergistic effect on doxorubicin cytotoxicity on MES-SA/Dx5 cells. Therefore, a combination of a nontoxic PARP1/ABCB1 inhibitor and doxorubicin treatment may present a promising strategy to combat multidrug resistant cancer.

## Data Availability

The data presented in this study are available in this article.

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
