# Peer review of "Olaparib Conjugates with Selenopheno[3,2-c]quinolinone Inhibit PARP1 and Reverse ABCB1-Related Multidrug Resistance"

_pharmaceutics, 2022, doi:10.3390/pharmaceutics14122571_

Round 1

Reviewer 1 Report

The authors reported the design and synthesis of 3 novel olaparib conjugates with selenophenoquinolinones, their ability to reverse doxorubicin resistance in uterus sarcoma cells as well as their mechanism of action. They found that the most potent chemosensitizer among studied compounds preserves PARP1 inhibitory activity and attenuates cells resistance to doxorubicin by inhibiting ABCB1 transporter activity.

Comments:

1. The authors should explain why the IC50 values of 5b and 5c in MES-SA/Dx-5 cells were less than those in MES-SA cells.

2. The authors explored the potential of the novel compounds to reverse resistance to doxorubicin. More P-gp substrates chemotheraeutic agents such as taxol, vincristine should also be tested.

3. It will be better if the authors detect the effects of the novel compounds in animal models.

4. "Statistical analysis" was missed in the "Materials and Methods".

Author Response

Dear Editor, Dear Referees,

We thank the editor for handling the manuscript. We also thank the referees for their efforts in reviewing the manuscript. The referees proposed constructive suggestions and gave overall positive comments regarding our manuscript. We have considered all the points made in peer review and a point-by-point response is detailed below.

Reviewer 1

  1. The authors should explain why the IC50 values of 5b and 5c in MES-SA/Dx-5 cells were less than those in MES-SA cells.
  • We propose that 5b is more cytotoxic to MES-SA/Dx-5 cells compared to its nonresistant counterpart in part because of ABCB1 transporter inhibition that is overexpressed in MES-SA/Dx-5 cells.
  • IC50 of 5c in MTT assay was relatively high (26.20±0.03), however, in Rho-123 accumulation assay lower concentrations were used (≤10 μM). No effect was observed for compound 5c at tested concentrations; however, it is possible that 5c affects ABCB1 transporter only at high concentrations. It is also possible that due to longer alkyl chain linker, 5c is a membrane-active compound and, thereby, potentiated doxorubicin activity. Nevertheless, as we have obtained more cytotoxic and more potent ABCB1 inhibitor 5b, compound 5c is less appealing.
  1. The authors explored the potential of the novel compounds to reverse resistance to doxorubicin. More P-gp substrates chemotheraeutic agents such as taxol, vincristine should also be tested.
  • This was a proof-of-concept study, more P-gp substrates will be evaluated in the future studies with the mechanism of action studied in detail when a compound with RF>10.
  1. It will be better if the authors detect the effects of the novel compounds in animal models.
  • The novel compounds will be studied in an animal model in the further research when the most suitable chemotherapeutic agent for co-treatment will be selected (with RF>10).
  1. "Statistical analysis" was missed in the "Materials and Methods".
  • Statistical analysis section was added to "Materials and Methods".

Best regards,

Pavel Arsenyan.

Reviewer 2 Report

In their paper, the authors reported the design and synthesis of 3 novel olaparib conjugates with selenophenoquinolinones, their ability to reverse doxorubicin resistance in uterus sarcoma cells as well as their mechanism of action. The PARPi is getting increasing attention in clinic because of its desired effectiveness in cancer treatment. The new conjugate introduced here is novel and has potential of clinical application. Overall, it is my impression that this paper could be an interesting contribution to Pharmaceutics pending the following points:

1.     Since the effect of PARP inhibition largely dependents on deficiency of HRR (Homologous Recombination Repair), which usually induced by BRCA1 or BRCA2 mutations, the authors should make a thorough description of mutation status of cell lines used in this study.

2.     Line 274: the IC50 value here was incorrect. Please check to make sure there are no similar errors.

Author Response

Dear Editor, Dear Referees,

We thank the editor for handling the manuscript. We also thank the referees for their efforts in reviewing the manuscript. The referees proposed constructive suggestions and gave overall positive comments regarding our manuscript. We have considered all the points made in peer review and a point-by-point response is detailed below.

Reviewer 2

  1. Since the effect of PARP inhibition largely dependents on deficiency of HRR (Homologous Recombination Repair), which usually induced by BRCA1 or BRCA2 mutations, the authors should make a thorough description of mutation status of cell lines used in this study.
  • MES-SA cell line does not have mutations in BRCA2 gene, BRCA1 gene alterations were not reported. https://doi.org/10.3390/cancers14051180
  • MCF-7 cell line has wild-type BRCA1 and BRCA2 genes, no mutations were found. doi:10.1038/sj.onc.1201752 ;     https://doi.org/10.4161/cbt.4.7.1909
  • No mutations in BRCA1/2 genes were reported for H9C2.
  • According to ATCC, HCC1937 has BRCA1 mutation (insertion C at nucleotide 5382). https://www.atcc.org/products/crl-2336
  • The text was corrected to prevent misinterpretation and to emphasize that only HCC1937 has BRCA1 mutation.
  1. Line 274: the IC50 value here was incorrect. Please check to make sure there are no similar errors.
  • Reported values were rechecked and corrected.

Best regards,

Pavel Arsenyan.